# Nutrigenomic Effect of Saturated and Unsaturated Long Chain Fatty Acids on Lipid-Related Genes in Goat Mammary Epithelial Cells: What Is the Role of PPARγ?

**DOI:** 10.3390/vetsci6020054

**Published:** 2019-06-11

**Authors:** Einar Vargas-Bello-Pérez, Wangsheng Zhao, Massimo Bionaz, Jun Luo, Juan J. Loor

**Affiliations:** 1Department of Veterinary and Animal Sciences, Faculty of Health and Medical Sciences, University of Copenhagen, Grønnegårdsvej 3, DK-1870 Frederiksberg C, Denmark; evargasb@sund.ku.dk; 2School of Life Science and Engineering, Southwest University of Science and Technology, Mianyang 621010, China; wangshengzhao01@163.com (W.Z.); luojun@nwafu.edu.cn (J.L.); 3Department of Animal Sciences and Division of Nutritional Sciences, University of Illinois Urbana-Champaign, Urbana, IL 61801, USA; 4Department of Animal and Rangeland Sciences, Oregon State University, Corvallis, OR 97331, USA

**Keywords:** nutrigenomics, LCFA, milk fat synthesis, peroxisome proliferator-activated receptor gamma (PPARγ), goat mammary epithelial cells

## Abstract

A prior study in bovine mammary (MACT) cells indicated that long-chain fatty acids (LCFA) C16:0 and C18:0, but not unsaturated LCFA, control transcription of milk fat-related genes partly via the activation of peroxisome proliferator-activated receptor gamma (PPARγ). However, in that study, the activation of PPARγ by LCFA was not demonstrated but only inferred. Prior data support a lower response of PPARγ to agonists in goat mammary cells compared to bovine mammary cells. The present study aimed to examine the hypothesis that LCFA alter the mRNA abundance of lipogenic genes in goat mammary epithelial cells (GMEC) at least in part via PPARγ. Triplicate cultures of GMEC were treated with a PPARγ agonist (rosiglitazone), a PPARγ inhibitor (GW9662), several LCFA (C16:0, C18:0, *t10,c12*-CLA, DHA, and EPA), or a combination of GW9662 with each LCFA. Transcription of 28 genes involved in milk fat synthesis was measured using RT-qPCR. The data indicated that a few measured genes were targets of PPARγ in GMEC (*SCD1*, *FASN*, and *NR1H3*) while more genes required a basal activation of PPARγ to be transcribed (e.g., *LPIN1*, *FABP3*, *LPL*, and *PPARG*). Among the tested LCFA, C16:0 had the strongest effect on upregulating transcription of measured genes followed by C18:0; however, for the latter most of the effect was via the activation of PPARγ. Unsaturated LCFA downregulated transcription of measured genes, with a lesser effect by *t10,c12*-CLA and a stronger effect by DHA and EPA; however, a basal activation of PPARγ was essential for the effect of *t10,c12*-CLA while the activation of PPARγ blocked the effect of DHA. The transcriptomic effect of EPA was independent from the activation of PPARγ. Data from the present study suggest that saturated LCFA, especially C18:0, can modulate milk fat synthesis partly via PPARγ in goats. The nutrigenomic effect of C16:0 is not via PPARγ but likely via unknown transcription factor(s) while PPARγ plays an indirect role on the nutrigenomic effect of polyunsaturated LCFA (PUFA) on milk fat related genes, particularly for CLA (permitting effect) and DHA (blocking effect).

## 1. Introduction

Fat is an important component of milk that accounts for many physical properties and manufacturing characteristics of milk and dairy products [1]. Milk fat is primarily composed of triacylglycerol (TAG) including fatty acids that originate from two sources: De novo synthesis within the mammary gland and the uptake of long-chain fatty acids (LCFA) from circulation [2].

Milk fat production is regulated by many factors including the availability of precursors and the abundance of proteins involved in milk fat synthesis, fatty acid (FA) activation, FA transport, FA desaturation, TAG synthesis, and milk fat globule formation and secretion [3,4]. A recent review of the literature highlighted a complex and still partially known network of transcription regulators responsible for the regulation of genes involved in milk fat synthesis [5].

In monogastric animals, LCFA can affect lipid and glucose metabolism via the activation of PPAR isotypes [6,7]. In bovine mammary tissue, gene network analysis revealed that the transcription of *PPARG* and its putative target genes was upregulated during lactation, suggesting a role for this nuclear receptor in the concerted regulation of genes that are responsible for the control of milk fat synthesis and secretion [4]. That led to subsequent work attempting to identify if LCFA of different chain lengths and degrees of unsaturation elicit distinct changes in putative target genes of PPARγ in bovine mammary epithelial cells [8]. Results of that work indicated that saturated LCFA, particularly C16:0 and C18:0, elicited the strongest effect on the transcription of measured genes and the response to these two LCFA was the closest to the ones observed with rosiglitazone, a known PPARγ agonist. Furthermore, in that work, the unsaturated LCFA had little effect on transcription of measured genes.

Goats produce milk with similar butterfat content compared with cows, but the FA profile is enriched with short-chain fatty acids, indicating a stronger contribution of de novo synthesis. Thus, the response observed in bovine cells might not necessarily be recapitulated in goats. Recently, the central role of PPARγ and its target genes was confirmed in the control of milk fat synthesis in goat mammary epithelial cells [9].

A more complete understanding at a molecular level (i.e., mRNA abundance) of the regulation of milk fat synthesis and secretion by LCFA would contribute to the development of nutrigenomics strategies to alter milk FA composition and optimizing milk fat production in dairy ruminants [10]. In non-ruminants, LCFA interact directly with transcription regulators such as PPAR, LXR, and hepatic nuclear factor (i.e., HFN-4α) to elicit a response [11]. At least in non-ruminants, the nuclear receptor PPARγ binds and is activated by LCFA, hence, it is amenable for fine-tuning milk fat synthesis.

Prior data indicated that saturated LCFA (C16:0 and C18:0) activated PPARγ by similarity in the transcriptomic response to rosiglitazone of several putative downstream PPARγ target genes to elicit some control of bovine milk fat synthesis [8]. However, in that study, the activation of PPARγ by LCFA was not demonstrated but only inferred. Furthermore, prior data were indicative of a lower response to rosiglitazone in goat mammary cells compared to bovine mammary cells [10,12]. Thus, the specific roles of LCFA in the regulation of milk fat synthesis via modulation of PPARγ in ruminant mammary cells and, more so, in goat mammary cells, remains unclear. The use of specific PPAR-isotype antagonists is an effective means to study the role of LCFA in activating PPAR [13]. Therefore, in the present study we aim to examine the hypothesis that LCFA alter the mRNA abundance of lipogenic genes in goat mammary epithelial cells (GMEC) at least in part via PPARγ using a combination of a synthetic specific antagonist of PPARγ with each LCFA, and to measure the mRNA abundance of lipogenic genes in primary GMEC.

## 2. Materials and Methods

### 2.1. Cell Culture and Treatments

All experiments utilized primary goat mammary epithelial cells (kindly provided by Peter Dovc, University of Ljubljana, Domzale, Slovenia). The GMEC cells were seeded in 75 cm^2^ flasks (430641, Corning, Glendale, AZ, USA), routinely cultured at 37 °C with 5% CO_2_, and grown as previously described [8].

To ensure a high degree of consistency in the initial conditions, subculture was performed several times to obtain a large number of cells. When the number of cells was enough to initiate the experiment, all the cells were split and pooled in a 50 mL sterile tube and mixed thoroughly before seeding in 6-well plates at a density of 20,000 cells/cm^2^. Cells remained in the growth medium for approximately 48 h (medium changed every 24 h). Once GMEC reached 80–90% confluence, the serum was removed and the GMEC were cultured in basal medium for 48 h followed by lactogenic medium for 24 h prior starting the experiment exactly as previously described [8]. Treated cells were incubated for 12 h and then harvested for RNA extraction. Each treatment was run in triplicate (intra- and inter-assay coefficient of variation <10%).

Seven treatments excluding a control media (NCTR) and a true control (i.e., with ethanol; PCTR) were used. Treatments were 50 μM of the positive activator of PPARγ Rosiglitazone (ROSI; 71740, Cayman Chemical, Ann Harbor, MI, USA) as previously done [8], 50 μM of the PPARγ inhibitor GW9662 (GW; 70785, Sigma-Aldrich, St. Louis, MO, USA), 100 μM of C16:0 (C16; N-16-A, Nu-Chek Prep Inc., Elysian, MN, USA), C18:0 (C18; N-18-A, Nu-Chek Prep Inc., Elysian, MN, USA), *t10,c12*-CLA (CLA; 1249, MATREYA LLC, State College, PA, USA), C22:6n-3 (DHA; 90310, Cayman Chemical), and C20:5n-3 (EPA; U-99-A, Nu-Chek Prep Inc). In addition, we used a combination of GW with each single LCFA (GWC16, GWC18, GWCLA, GWDHA, and GWEPA). The selection of the two saturated LCFA and doses was based on the previously reported strong effects on PPAR in bovine [12]. The selection of the unsaturated LCFA was based on their known role in milk fat synthesis, mostly negative [14].

### 2.2. RNA Extraction, RNA Quality, Primer Design, and qPCR

Total RNA from cells was extracted with QIAzol Lysis Reagent (QIAGEN, Valencia, CA, USA) as previously described [15]. Genomic DNA was removed with DNase I using RNeasy Mini Kit columns (Qiagen, Valencia, CA, USA). RNA concentration and purity were determined with a NanoDrop ND-1000 spectrophotometer (NanoDrop Technologies, Wilmington, DE, USA). The purity of RNA (A260/A280) for all samples was greater than 1.9. The RNA integrity was assessed using a 2100 Bioanalyzer (Agilent Technologies, Santa Clara, CA, USA). All samples had RNA integrity number ≥5 (median of 8), adequate for RT-qPCR analysis [16]. A portion of the assessed RNA was diluted to 100 ng/μL using DNase-RNase free water prior to reverse transcription. Abundant cDNA was prepared to detect all selected genes. Each cDNA was synthesized by reverse transcription PCR and used for qPCR as previously described [15].

Primer Express 3.0 software (Applied Biosystems, Foster City, CA, USA) was employed to design primers as previously described [15]. Primers were designed to fall across exon-exon boundaries when possible to avoid amplification of genomic DNA. The exon junctions were unveiled by blasting the sequence available in the National Center of Biotechnology Information against sheep genome in the UCSC Genome Browser. Primer features and gene description are available in Appendix A, respectively. Primers were validated as previously described [15].

For qPCR, each sample was run in triplicate, and a 6-point relative standard curve plus the non-template control were used. The 4-fold-dilution standard curve was made using cDNA from RNA pool of all the samples. The reactions were performed in a MicroAmp™ Optical 384-Well Reaction Plate (Applied Biosystems) using the following conditions: 2 min at 50 °C, 10 min at 95 °C, 40 cycles of 15 s at 95 °C (denaturation), and 1 min at 60 °C (annealing and extension). The presence of a single PCR product was verified by the dissociation protocol using incremental temperatures to 95 °C for 15 s plus 65 °C for 15 s. Final data were calculated with the 7900 HT Sequence Detection Systems Software (version 2.4, Applied Biosystems).

### 2.3. Data Transformation, Relative mRNA Abundance, and Statistical Analysis

*GAPDH*, *RPS9*, and *UXT*, which have been used as suitable internal controls with mammary tissue or cells [8,15], were tested as internal control genes for the present experiment. Data were analyzed by geNorm algorithm [17]. All three genes had an M-value of <0.27 and the use of the three genes provided a V-value of 0.095, indicating a very good normalization. Data were normalized using the geometrical mean of *GAPDH*, *RPS9*, and *UXT*.

Final data were transformed to determine the expression ratio relative to the mean of the PCTR for each gene. Data were then log_2_ transformed prior statistical analysis. Outliers were checked using Proc Reg of SAS (v.9.3, SAS Institute Inc., Cary, NC, USA) removing data with a studentized t > 3.0 prior statistical analysis. Due to the combination of treatments, three separated statistical analyses were performed using the GLM procedure of SAS with treatment as main effect: (1) To assess the effect of each treatment in the absence of the GW9662 (i.e., NCTR, PCTR, ROSI, C16, C18, CLA, DHA, and EPA); (2) to compare the effect of each combination of LCFA with GW (i.e., compared were GW, GWC16, GWC18, GWCLA, GWDHA, and GWEPA); and (3) to assess the direct comparison between LCFA and the same LCFA with GW (i.e., PCTR vs. GW, C16 vs. GWC16, C18 vs. GWC18, CLA vs. GWCLA, DHA vs. GWDHA, and EPA vs. GWEPA). Significance was declared with a *p*-value ≤ 0.05 and tendency with a *p*-value between 0.05 and 0.10. Correlation analysis was performed using Proc Corr of SAS (v9.4). Data from reference [8] were correlated with data from common genes measured in the present experiment and used as fold change relative to control. The relative mRNA abundance of all measured genes was calculated as previously reported [4]; that is, 1/E(ΔCt), where E is the efficiency of PCR amplification (E = 10^–1/–log curve slope^), and ΔCt is calculated as [Ct gene – (geometrical mean Ct of 3 reference genes)].

## 3. Results

### 3.1. Relative Abundance of Measured Transcripts

Among measured transcripts (Appendix A), the Spot 14-Related Protein (*MID1IP1*) and the key de novo synthesis *ACACA* were the most abundant genes followed by several genes coding for transcription factors such as *RXRA* and *NCOR1*, and metabolism-related genes, such as *OXCT1* and *PLIN2*. Very low relative abundance was detected for *PPARG* and *PPARD* (<0.2%). Among other transcripts involved in fatty acid and triacylglycerol synthesis, genes coding for desaturases and *FASN* were ca. 1% with other genes being <2% abundance. Very low abundance, except for *VLDLR* with >2%, was detected for transcripts involved in fatty acid transport (<0.1%) with *LPL* being barely detectable (0.003% with a Ct value of 32.2).

### 3.2. Genes Affected by Synthetic Agonist and Antagonist of PPARγ

#### 3.2.1. LCFA and Triacylglycerol Synthesis

The activation of PPARγ by rosiglitazone increased the transcription of *FASN*, *SCD1*, and *AGPAT6* (Figure 1). Treatment of cells with the PPARγ antagonist GW9662 decreased the basal transcription of *ACACA* (*p* = 0.09), *LPIN1*, *GPAM*, *DGAT1* (*p* = 0.06), and *ACSL1* and increased the transcription of *FASN* (Figure 1).

#### 3.2.2. LCFA Transport

Rosiglitazone affected none of the transcripts related to LCFA transport, while the basal transcription of *LPL*, *SLC27A1*, and *FABP3* was decreased by GW9662 (Figure 2).

#### 3.2.3. Transcription Regulation and other Functions

Among transcription regulators, ROSI treatment increased the mRNA abundance of *NR1H3* while the inhibition of PPARγ decreased the basal mRNA abundance of *SREBF2*, *NR1H3*, *NCOR1*, *PPARG*, and *PPARD* (Figure 3). Of other transcripts, ROSI only increased the transcription of *PLIN2* (Figure 4).

A summary of transcripts that can be considered PPARγ targets based on the above data is provided in Table 1. Overall, under the conditions of this study, most of the genes associated with LCFA synthesis and desaturation were either direct PPARγ targets that could be modulated by a synthetic agonist (e.g., *SCD1* and *FASN*) or their basal expression was controlled by PPARγ but their modulation could be PPARγ-independent (*ACACA*, *ACSL1*, *DGAT1*, *GPAM*, and *LPIN1*). Among the latter, transcription of *ACACA* and *DGAT1* appears to be less dependent from PPARγ activation.

Among LCFA transporters, only *LPL*, *FABP3*, and *SLC27A1* are likely PPARγ targets; however, PPARγ activation appears to be essential only for their basal transcription. Similarly, for many of the mRNA measured related to transcription regulation, basal PPARγ activation was essential for their transcription including *NCOR1*, *PPARD*, *PPARG*, and *SREBF2*, while the only true PPARγ target gene (i.e., its transcription is tightly regulated by PPARγ) appeared to be *NR1H3*. Among other transcripts measured, only *PLIN2* was influenced by PPARγ modulation, although the mechanism is unclear.

A correlation analysis of the response of all LCFA with rosiglitazone is available in Appendix A. The data indicated that C16:0 and C18:0 had a significant positive correlation with ROSI while the response to unsaturated LCFA was not correlated to ROSI.

### 3.3. mRNA Abundance and Overall Effects of LCFA

#### 3.3.1. Fatty Acid Synthesis and Desaturation

The saturated LCFA, C16:0, increased the transcription of *SCD1*, *FADS1*, and *LPIN1*. However, the effect was independent of PPARγ since the addition of GW did not affect the transcriptional response of these genes to C16:0. The inhibition of PPARγ decreased but did not nullify the positive effect of C16:0 on the transcription of *FASN* and allowed the increase in transcription of *AGPAT6*. C18:0 increased the transcription of *SCD1*, *FASN*, *FADS1*, and *AGPAT6*. However, for the former three, the effect was nullified when PPARγ was inhibited (Figure 1).

Among unsaturated LCFA, CLA inhibited the transcription of *FADS1* and *LPIN1*, but the effect disappeared when PPARγ was inhibited. Furthermore, CLA induced the transcription of *AGPAT6* only when GW9662 was used. Except for the transcription of *AGPAT6* that was upregulated by DHA in a PPARγ-independent fashion, DHA had a negative effect on the transcription of all the genes measured in this category (with a tendency for *DGAT1*) only when PPARγ was inhibited. Except for *AGPAT6*, where the increase in transcription was achieved only when PPARγ was inhibited, and *DGAT1* that was not affected, EPA downregulated the transcription of all measured genes in this category independently from PPARγ. However, upon addition of GW9662, EPA did not affect transcription of *ACSL1*.

#### 3.3.2. Fatty Acid Transport

Regardless of the presence of GW9662, all LCFA upregulated transcription of *CD36* but the effect was abolished for C18:0 and CLA when PPARγ was inhibited (Figure 2). Among saturated LCFA, C16:0 increased mRNA abundance of *LPL*, *FABP3*, and *FABP4*. The effect disappeared for transcription of *LPL* when GW9662 was added. C18:0 increased the transcription of *FABP3* and *FABP4* only when PPARγ was not inhibited. Among unsaturated LCFA, CLA only decreased transcription of *FABP3* and increased mRNA abundance of *FABP4* but the effect disappeared when GW9662 was added. The DHA reduced the transcription of *VLDLR*, *SLC27A1*, and *FABP3* (*p* = 0.10) and the transcription of *FABP4* was significantly reduced when GW9662 was present. EPA decreased the transcription of *FABP4* independently from inhibition of PPARγ. Despite not being different compared with the controls, EPA partly counteracted the inhibition of *LPL* transcription by GW9662.

#### 3.3.3. Transcriptional Regulation of Lipogenesis

The saturated LCFA C16:0 and C18:0 upregulated transcription of *SREBF1* and *INSIG1* and downregulated transcription of *NCOR1* independently from the presence of the PPARγ inhibitor (Figure 3). However, C18:0 did not decrease mRNA abundance of *NCOR1* when GW9662 was present. C16:0 increased transcription of *PPARG*. C18:0 nullified the downregulation of *NCOR1* and *PPARD* by GW9662. CLA decreased transcription of *SREBF1*, *INSIG1*, *RXRA*, and *NCOR1* only when GW9662 was not present. DHA decreased mRNA abundance of *SREBF1*, *SREBF2*, *SCAP*, and *NR1H3* and tended to decrease the transcription of *NCOR1* (*p* = 0.09) and *PPARD* (*p* = 0.06) only when PPARγ was inhibited. EPA decreased transcription of *SCAP* and *NR1H3* independently from PPARγ inhibition.

#### 3.3.4. Lipid Droplet Formation and Ketone Body Utilization

Transcription of *PLIN2* was increased by all treatments. PPARγ inhibition significantly increased the effect of EPA, hampered the effect of C18:0, and tended to decrease the magnitude of effect of CLA on *PLIN2* transcription (Figure 4). The transcription of *OXCT1* was downregulated by EPA in a PPARγ-independent fashion and was numerically (*p* = 0.11) downregulated by DHA only when PPARγ was inhibited. The transcription of 3-Hydroxybutyrate Dehydrogenase 1 (*BDH1*) was negatively affected by CLA but this effect disappeared when PPARγ was inhibited. For *MID1IP1*, C16:0 increased its mRNA abundance independently from PPARγ inhibition while the increase of transcription by EPA was evident only when PPARγ was inhibited.

### 3.4. Transcriptomic Effect of LCFA via PPARγ

A summary of the effect of each fatty acid via PPARγ is reported in Table 2. Transcription of *FASN*, *LPL*, and *SREBF1* was affected by C16:0 via PPARγ. Expression of a larger number of genes was affected by C18:0 at the least in part via PPARγ compared to C16:0. Those included *FADS1*, *FASN*, *SCD1*, *CD36*, *FABP3*, *FABP4*, and *PLIN2*. None of the measured transcripts were upregulated by unsaturated LCFA via PPARγ. However, several transcripts were downregulated by CLA only in the absence of GW9662 while most of the measured transcripts were downregulated by DHA only when GW9662 was present.

A summary of the effect on expression of measured genes and the likely role of PPARγ in such an effect is summarized for saturated and unsaturated LCFA in Figure 5 and Figure 6, respectively. Our data indicated that C18:0 was modulating PPARγ more than C16:0, with only 4 vs. 8 genes, whose expression was clearly controlled by C16:0 and C18:0 via PPARγ, respectively (Figure 5). Despite none of the transcripts measured were affected by unsaturated LCFA directly through PPARγ, most of them were affected by CLA and DHA (and several affected by EPA) with an indirect influence of PPARγ. Nine transcripts were downregulated and 1 upregulated by CLA through unknown TF that appeared to be dependent on basal activation of PPARγ while CLA increased the transcription of 6 genes via unknown TFs that are inhibited by PPARγ (Figure 6). For DHA, the effect was even more dramatic. Downregulation of the transcription by DHA of 16 out of 17 genes measured in our experiment was via one of more TFs that are inhibited by PPARγ activation (Figure 6). For EPA, the interaction was more complex. Transcription of 4 genes was modulated by EPA through one or more TFs that are dependent on basal activation of PPARγ, while the transcription of 6 genes was modulated by EPA via one or more TF that are inhibited by PPARγ (Figure 6).

### 3.5. Comparison with MACT Cells

#### 3.5.1. Proportion of mRNA Abundance of Lipogenic Genes between MACT and GMEC

A direct comparison on % transcript abundance of commonly measured transcripts coding for lipogenic proteins in GMEC used in the present experiment and immortalized bovine mammary alveolar cells (MACT) [8] is available in Appendix A. Data indicated that in both cell types the proportion of transcripts in each functional category was very similar. The abundance of transcripts of measured genes coding for proteins involved in fatty acid transport accounted for a minor part (4.65% and 0.65% in MACT and GMEC, respectively). The % mRNA abundance of genes involved in fatty acid and triacylglycerol synthesis accounted for the large majority (67.7% and 66.9% in MACT and GMEC, respectively) followed by mRNA of transcription regulators (27.7% and 32.4%, in MACT and GMEC, respectively).

Despite an overall similarity in abundance of mRNA coding for proteins involved in specific functions, we detected differences in specific genes. MACT cells had a higher proportion of *FABP3* compared to GMEC. The larger difference in % mRNA abundance between the two cell types was detected for transcripts related to fatty acid and triacylglycerol synthesis, with a larger proportion of *ACACA* (50.5% vs. 1.1%) and *LPIN1* (6.3% vs. 2.7%) in GMEC vs. MACT while mRNA of *FASN*, *SCD1*, *AGPAT6*, and *DGAT1* was proportionally more abundant in MACT cells (Appendix A). Among transcription regulators, the proportion of mRNA was somewhat conserved between the two cells; however, GMEC presented a higher mRNA abundance of *SREBF2* and a lower proportion of sterol regulatory element binding protein 1 (SREBP1) co-regulators *SCAP1* and *INSIG1* compared with MACT cells (Appendix A).

#### 3.5.2. C18:0 Affects Transcription of More Lipogenic Genes in GMEC than MACT

The data produced in the present study can be compared to the work from reference [8] where MACT cells were used with the same experimental set up as the present experiment, with exclusion of the GW9662, which was used only in the present experiment. Results of the comparison are available in Appendix A.

Compared with MACT cells, GMEC had only three transcripts affected by rosiglitazone. Two transcripts affected by rosiglitazone were common with MACT (*FASN* and *AGPAT6*), but with less magnitude of change in GMEC compared with MACT.

Transcription of more genes was affected by C16:0 in GMEC compared with MACT (9 vs. 7), with four transcripts (*FABP3*, *FABP4*, *SCD1*, and *INSIG1*) commonly affected in both cell types. Eight transcripts were affected by C18:0 in both cell types. Among them, five were similarly affected but transcription of *FASN* was upregulated in GMEC and downregulated in MACT, with also the transcription of the de novo gene *ACACA* downregulated in MACT while not affected in GMEC.

Among unsaturated LCFA, CLA affected transcription of more genes in MACT compared with GMEC with only three transcripts similarly affected (*FABP3*, *SREBF1*, and *INSIG1*, all downregulated). Likewise, EPA affected transcription of more genes in MACT vs. GMEC (8 vs. 7) with transcription of *SCD1* and *SCAP* being similarly downregulated by EPA in both cell types, but transcription of *INSIG1* being downregulated in GMEC but upregulated in MACT.

The response of measured lipogenic-related genes to LCFA in GMEC and MACT was significantly correlated for C16:0, CLA, and EPA, but was not correlated for the response to C18:0 (Appendix A). The response to C18:0 in MACT was more correlated with the response to CLA and EPA than the response to C16:0 in GMEC.

## 4. Discussion

### 4.1. GMEC Have a Weak Response to Rosiglitazone but Respond to PPARγ Antagonist

Compared with bovine mammary cells, goat mammary cells appear less responsive to rosiglitazone. This was previously demonstrated using gene reporter data [10] and supported by the lower number and magnitude of transcripts affected by rosiglitazone in our experiment compared with MACT cells [8]. It is noteworthy that the protein sequence of PPARγ is highly conserved between bovine and goat, considering that 99.6% of amino acid sequence is identical when compared using Pairwise Sequence Alignment (LALIGN) [18] and the ligand-binding domain has 100% conservation (Appendix A). The reason for the observed weaker response to rosiglitazone in GMEC vs. MACT is unclear. In this experiment, we used 50 μM of rosiglitazone. This is a very large dose when compared with the dose that activates PPARγ in primates [19]. That dose appeared to be effective in activating PPARγ in bovine cells, as indicated above. The weak responses to rosiglitazone in GMEC can indicate that PPARγ is not a major player in GMEC, which would contradict a large number of publications supporting a role of PPARγ in controlling transcription of milk fat synthesis-related genes in goats [10,20,21,22,23,24]. However, the larger abundance of *FABP3* in MACT vs. GMEC could be one of the causes. This transcript was originally denominated Mammary-Derived Growth Inhibitor due to its large abundance in terminal differentiated mammary tissue [25]. This is also one of the most abundant transcripts in bovine mammary tissue and it is involved in intracellular transport of LCFA and PPAR activation [4,26]. Taken together, our data suggests that when PPARγ is stimulated it affects mammary lipogenesis by controlling expression of several key lipogenic genes (in particular *FASN* and *SCD1*) in GMEC. Our findings are also supported by prior data in GMEC [9].

The compound GW9662 is a potent and selective PPARγ antagonist [27] with inhibition in human PPARγ at doses < 10 nM; however, in humans, higher doses also inhibit PPARα (between 100 and 1000 nM) and PPARβ/δ (>1000 nM). In our experiment, the 50 μM would likely guarantee a complete inhibition of PPARγ but could also have inhibited the other two PPAR isotypes. Without more sophisticated molecular biology techniques, it is not possible to determine if GW9662 is as effective in inhibiting the various PPAR isotypes in GMEC as it is in primate cells. This is a limitation of the present study and was taken into account when considering the results with the LCFA in combination with GW9662.

### 4.2. C16:0 Elicits a Strong Transcriptome Effect but C18:0 Is a More Specific PPARγ Agonist in GMEC

As observed previously [8], C16:0 had the strongest effect among treatments on the transcription of measured lipogenic genes. A role for C16:0 in regulating de novo FA synthesis was reported earlier in bovine and ovine dispersed mammary epithelial cells when exogenous C16:0 led to increased synthesis and incorporation of butyrate into TAG [28]. Because C16:0 is the main product of de novo FA synthesis in mammary cells, it was previously proposed that C16:0 could serve as a feed-forward mechanism for copious milk fat synthesis during the normal course of lactation [4]. There is, however, also the possibility that C16:0 regulates milk fat synthesis via PPARγ [10]. Based on our data, except for *LPL*, *FASN*, and *SREBF1*, none of the effects of C16:0 on the transcription of lipogenic genes appeared to be via PPARγ (or other PPAR isotypes). *LPL*, *FASN*, and *SREBF1* genes are, however, key for milk fat synthesis, specifically to import preformed LCFA via hydrolysis of TAG from VLDL and for de novo synthesis [4]. Thus, the data clearly indicated that C16:0 can regulate milk fat synthesis at a transcriptomic level, but the majority of the effect is likely not by activating PPARγ. The data indicated that other TF should be investigated to fully understand the transcriptomic effect of C16:0 on milk fat synthesis in goats.

Most of the transcriptomic effect of C18:0 on lipogenic genes measured is instead via PPARγ (or other PPAR isotypes) in GMEC. As for C16:0, C18:0 also appears to be bioactive in bovine mammary gland increasing de novo synthesis of LCFA [29]. Activation of PPARs by C18:0 was also suggested from previous work [8]; however, in that study C16:0 among the two saturated LCFA appeared to be the strongest PPAR agonist. In our study, the response of C18:0 was more similar to rosiglitazone and the use of the PPARγ inhibitor was indicative of a more specific activation of PPARs (likely PPARγ) by C18:0 compared with C16:0. The two saturated LCFA are both weak activators of PPARα in mouse, but are both activators of PPARα in human [30]. Taken together, the data are indicative of C18:0 being a more specific PPARγ agonist in GMEC than C16:0. Considering MACT cell data discussed above, the present study indicates species-specific effects of the two saturated LCFA on PPAR.

### 4.3. Basal Activation of PPAR Is Essential for Nutrigenomic Activity of CLA and DHA

Unsaturated LCFA are potent PPAR agonists in non-ruminants, specifically in mouse [30,31], while they have a very low agonistic effect towards bovine PPAR isotypes [10]. The response observed in bovine cells was confirmed in GMEC in the present experiment.

The unsaturated LCFA are known to decrease abundance of lipogenic genes in bovine [10,32]. In our experiment, all three unsaturated LCFA decreased mRNA abundance of several of the measured genes. However, due to the experimental set-up (i.e., the use of the PPARγ antagonist) a novel and unanticipated effect was detected for CLA and DHA. The DHA decreased the abundance of milk fat synthesis-related genes in GMEC only in the absence of a basic activation of PPARγ, while CLA repressed the transcription of lipogenic genes only when basic PPARγ activation was present. Therefore, for DHA the data suggest that PPARγ was likely blocking the expression/activity of another (or others) TF that respond/s to DHA repressing the expression of milk fat synthesis genes. Considering that PPARs are known to act as transrepressors for several transcription factors, this idea is highly likely [33]. In contrast, our data indicated that a TF that binds CLA and represses the transcription of lipogenic genes appears to be under the control of PPARγ. Those findings are hard to explain and, based on the data available, we can only speculate about a possible explanation.

Although none directly bind LCFA, some polyunsaturated LCFA (PUFA) inhibit lipogenic genes in rodent liver via several TF such as SREBP1 [34]. A possible TF that is known to be inhibited by PUFA and interacts with PPAR is the Carbohydrate Response Element Binding Protein (ChREBP). It has been demonstrated that PUFA inhibit de novo lipogenesis in liver of mouse via inhibition of ChREBP [35]. In murine brown adipose tissue, the activation of PPARα partially suppresses ChREBP activity [36]. There are no data on the interaction between PPAR isotypes and ChREBP in ruminants. The expression of ChREBP is decreased in bovine mammary by CLA [37]. Thus, it is possible that ChREBP plays a role in the inhibition of lipogenic genes by CLA in GMEC. This would even be more so if ChREBP expression or activity is under control of PPARγ; however, in mouse adipocytes, ChREBP controls the activity of PPARγ [38] but there is no evidence of the opposite. Thus, the role of ChREBP and the interaction with PPAR isotypes does not fully support the observed inhibition of lipogenic genes by CLA only when basic PPARγ activation is present.

For the effect of DHA, a plausible candidate is the nuclear factor kappa-light-chain-enhancer of activated B cells (NFκB). This is a nuclear receptor essential for the inflammatory response and its activity is directly repressed by PPARγ via modulation of the transcription of IκBα, as observed in mouse [39]. Similarly, a previous study [40] also reported that DHA decreases the mRNA levels of TNF-α, IL-6 and IL-1β and suppresses NFκB activation in lipopolysaccharide-stimulated bovine mammary epithelial cells. The authors attributed the anti-inflammatory mechanism of DHA to its ability to activate PPARγ. It is noteworthy in this regard that NFκB is an upstream regulator of several of the lipogenic genes repressed by DHA in the present experiment including *FABP3*, *LPIN1*, and *FASN*, as revealed by an in silico analysis [5]. The NFκB is known to be inhibited by omega-3 LCFA, including DHA [41]. Thus, it is possible that several of the lipogenic genes in GMEC are under the control of both PPARγ and NFκB.

The regulation of the transcription of genes involved in milk fat synthesis is likely complex, involving many transcription factors [5]. For this reason, it is also possible that PPARγ represses still unknown TF that, upon binding to DHA, suppresses the transcription of lipogenic genes in GMEC by binding a PUFA-RE, as also theorized several years ago [42].

The downregulation of most lipogenic genes measured by EPA seems to agree in part with a previous in vivo study [43]. In that study, a duodenal infusion of fish oil with high EPA content resulted in lower milk fat concentration. The lack of effect of EPA on PPARγ is in agreement with a previous study in bovine mammary cells [8]; however, in non-ruminants, EPA activate PPARγ as observed in mouse myoblasts [44] and hepatocytes [45]. These data highlight the previously proposed differences between ruminants and non-ruminants in the response of PPAR to PUFA [12].

### 4.4. Response of Mammary Cells from Goat vs. Bovine to LCFA: Similar but Different

The differences observed between GMEC and MACT in the response to rosiglitazone might be attributed to interspecies differences and the origin of the cells used for each study (i.e., primary vs. immortalized). As mentioned in reference [22], a challenge/bias when comparing in vitro data among studies is the wide range of cell culture conditions (e.g., culture medium, absence/presence of prolactin, and different laboratory protocols among other variables). Although in both studies the culture conditions were almost identical, the two studies were carried out more than five years apart. On the other hand, in reference [46], differences on milk fat composition in cows and goats in the response to sunflower oil were suggestive of interspecies differences in the control of milk fat synthesis. For instance, in that study, a lower sensitivity to the inhibitory effects of *t10,c12*-CLA were observed in goats compared with cows. Part of the difference was attributed by the authors to ruminal lipid metabolism with less pronounced shift in the biohydrogenation pathways that generate LCFA with a trans-10 double bond in goat vs. cow [46]. The high similarity in the response to CLA and EPA between GMEC and MACT, is indicative of a high conservation of the milk fat depressing response to PUFA between the two species. The larger difference observed between GMEC and MACT was in the response to C18:0, with a stronger transcriptomic response in goats vs. bovine.

Collectively, our data revealed that GMEC have a lower sensitivity to rosiglitazone but LCFA control expression of lipogenic genes similarly to bovine MACT. As for MACT, part of the effect of LCFA on expression of genes related to lipid metabolism was via activation of PPARγ.

### 4.5. Limitations of the Study

Our study presents several limitations. The findings would have been strengthened by the use of biological replicates beside the use of technical replicates. The use of gene expression to determine if a PPAR isotype is activated is also problematic. Although several PPAR isotype-specific target genes have been proposed previously for ruminants [12], most of those were not demonstrated but mainly based on gene expression data obtained in bovine combined with prior data obtained from non-ruminants using more precise molecular techniques, such as gene reporter assays [10]. Furthermore, transcription of each gene is under control of several transcription factors. Therefore, findings from the present work need to be validated using molecular techniques that allow the study of each specific PPAR isotype. Unfortunately, such techniques are not yet available for ruminants.

## 5. Conclusions

Despite the above limitations, we can conclude that treatment with GW allowed the uncovering that a minimum activation of PPARγ is essential for the transcription of several genes involved in FA synthesis (*LPIN1*, *GPAM*, *DGAT1*, and *ACSL1*), FA transport (*LPL*, *SLC27A1*, and *FABP3*), and regulation of lipogenic genes transcription (*SREBF2*, *NR1H3*, *NCOR1*, and *PPARG*). Furthermore, treatment with rosiglitazone allowed identifying *FASN*, *SCD1*, *NR1H3*, and *PLIN2* as genes which transcription can be increased via PPARγ in GMEC. Interestingly, very few genes were true PPARγ targets; however, for most of the genes, the activity of PPARγ appeared to be essential or important for their transcription; thus, they might be a PPARγ target but their transcription was not increased via nutrigenomic approaches in goats.

Results suggest that, like bovine, C16:0 and C18:0 are strong activators of the transcription of milk fat synthesis-related genes in GMEC while CLA, DHA, and EPA inhibit the transcription of many of them. Moreover, the negative effect of CLA on transcription of lipogenic genes requires a basal activation of PPAR (likely PPARγ) while the downregulation of lipogenic genes by DHA only happens when PPARγ is deactivated. Effect on transcription of lipogenic genes by EPA is not at all related to PPARγ. It is important to mention that many of the transcriptomic effects of LCFA observed were independent from PPARγ suggesting the involvement of other transcriptional factors. Those results support the use of saturated LCFA, especially C18:0, to improve milk fat synthesis via nutrigenomic intervention in dairy goats.

## Figures and Tables

**Figure 1 vetsci-06-00054-f001:**
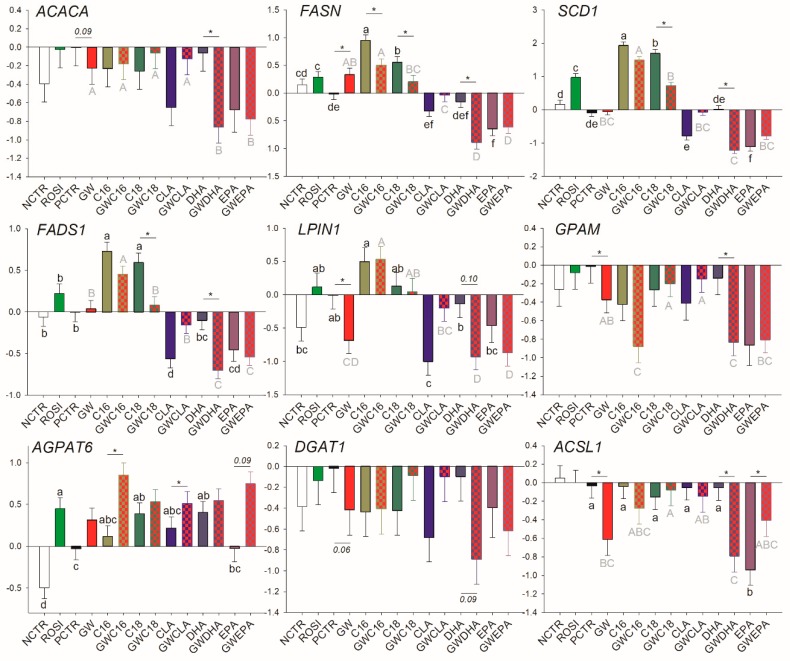
Effect of long-chain fatty acids (LCFA; palmitic acid [C16], stearic acid [C18], trans10, cis12 conjugated linoleic acid [CLA], docosahexaenoic acid [DHA], and eicosapentaenoic acid [EPA]), peroxisome proliferator-activated receptor gamma (PPARγ) agonist (ROSI; rosiglitazone) and LCFA + PPARγ antagonist (GW; GW9662) on transcription of genes related to fatty acid synthesis and desaturation and triglycerides synthesis. Lower case letters and uppercase letters denote statistical differences (*p* < 0.05) between treatments without GW and treatments with GW, respectively. * denotes statistical difference (*p* < 0.05) between presence and absence of GW in the same LCFA treatment. For GW treatment alone, * denotes statistical difference (*p* < 0.05) with the PCTR.

**Figure 2 vetsci-06-00054-f002:**
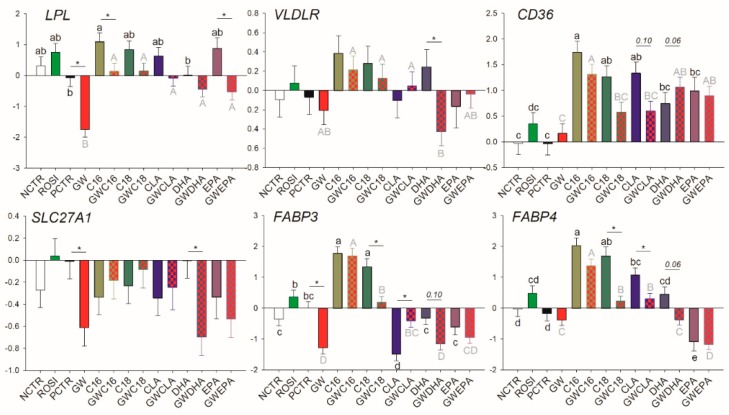
Effect of LCFA (palmitic acid [C16], stearic acid [C18], trans10, cis12 conjugated linoleic acid [CLA], docosahexaenoic acid [DHA], and eicosapentaenoic acid [EPA]), PPARγ agonist (ROSI; rosiglitazone) and LCFA + PPARγ antagonist (GW; GW9662) on transcription of genes involved in LCFA transport. Lower case letters and uppercase letters denote statistical differences (*p* < 0.05) between treatments without GW and treatments with GW, respectively. * denotes statistical difference (*p* < 0.05) between presence and absence of GW in the same LCFA treatment. For GW treatment alone, * denotes statistical difference (*p* < 0.05) with the PCTR.

**Figure 3 vetsci-06-00054-f003:**
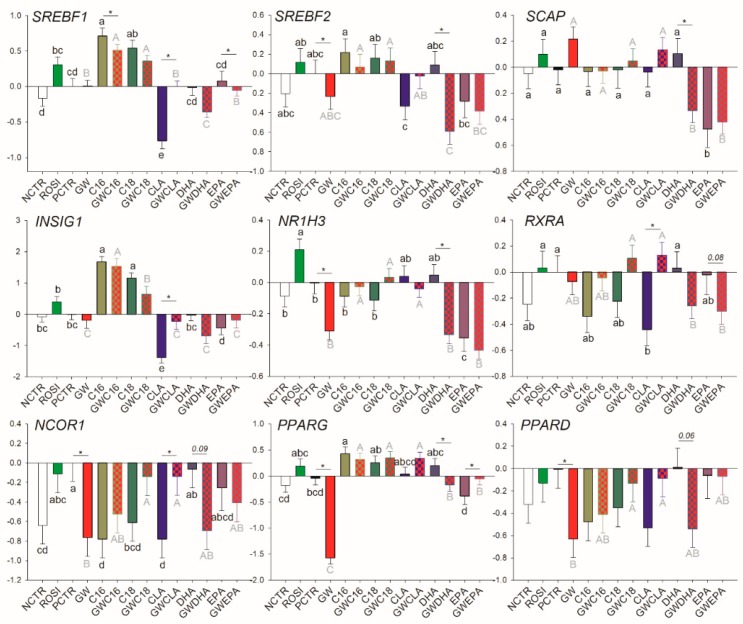
Effect of LCFA (palmitic acid [C16], stearic acid [C18], trans10, cis12 conjugated linoleic acid [CLA], docosahexaenoic acid [DHA], and eicosapentaenoic acid [EPA]), PPARγ agonist (ROSI; rosiglitazone) and LCFA + PPARγ antagonist (GW; GW9662) on transcription of genes involved in transcriptional regulation of lipogenesis. Lower case letters and uppercase letters denote statistical differences (*p* < 0.05) between treatments without GW and treatments with GW, respectively. * denotes statistical difference (*p* < 0.05) between presence and absence of GW in the same LCFA treatment. For GW treatment alone, * denotes statistical difference (*p* < 0.05) with the PCTR.

**Figure 4 vetsci-06-00054-f004:**
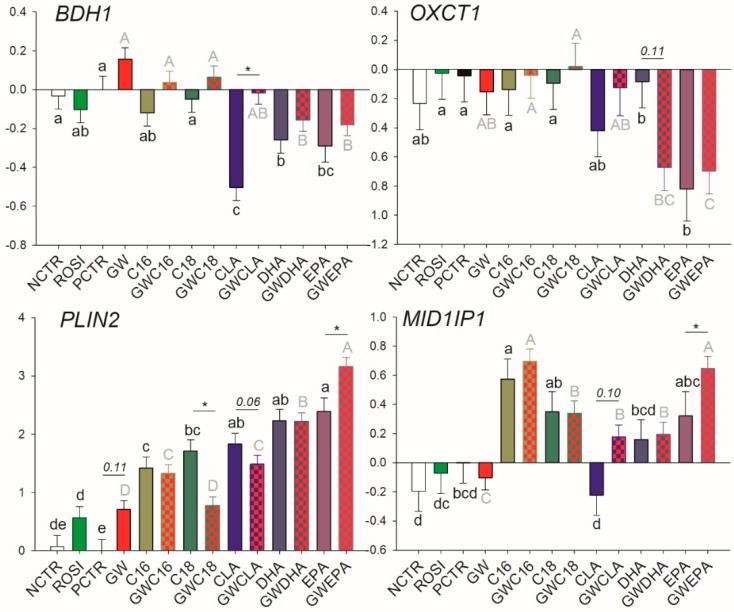
Effect of LCFA (palmitic acid [C16], stearic acid [C18], trans10, cis12 conjugated linoleic acid [CLA], docosahexaenoic acid [DHA], and eicosapentaenoic acid [EPA]), PPARγ agonist (ROSI; rosiglitazone) and LCFA + PPARγ antagonist (GW; GW9662) on transcription of genes involved in lipid droplet formation and ketone body utilization. Lower case letters and uppercase letters denote statistical differences (*p* < 0.05) between treatments without GW and treatments with GW, respectively. * denotes statistical difference (*p* < 0.05) between presence and absence of GW in the same LCFA treatment. For GW treatment alone, * denotes statistical difference (*p* < 0.05) with PCTR.

**Figure 5 vetsci-06-00054-f005:**
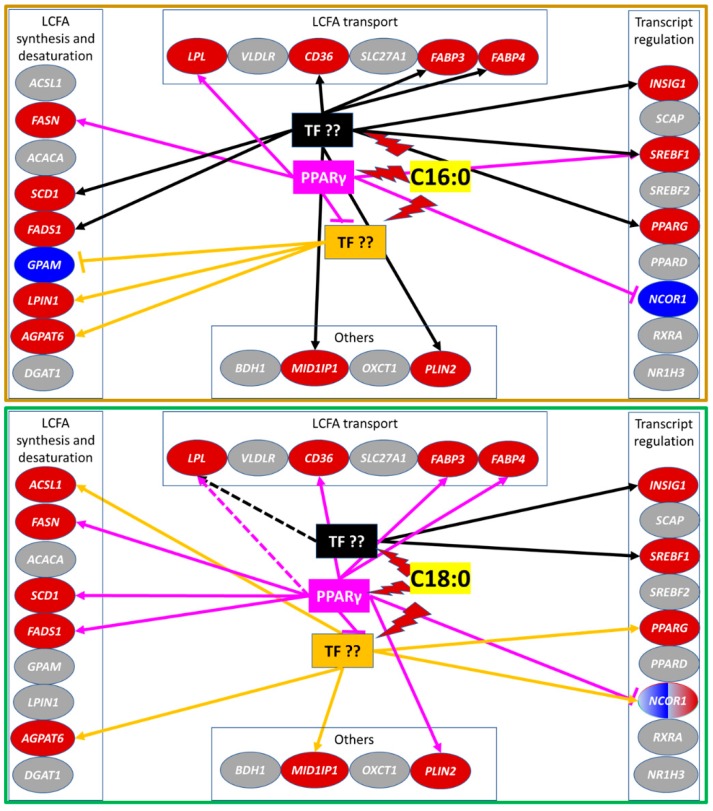
Schematic summary of the results from the present experiment for the response of transcription of various genes to saturated LCFA C16:0 and C18:0 clustered in functional groups related to milk fat synthesis. It is also reported the possible interaction of LCFA with PPARγ or other unidentified transcription factors (TF) to elicit the observed responses. Grey shade denotes no significant change, red shade denotes upregulation, and blue shade denotes downregulation relative to control. In connecting lines, arrow head denotes induction while flat head denotes inhibition. Lightning symbol denotes induction of TF by the LCFA.Based on the results from our experiment, C16:0 controlled the expression of 4 genes (*SREBF1*, *FASN*, *LPL*, and *NCOR1*) via PPARγ or partly via this transcription factor (TF) (as was the case for *SREBF1*). The expression of other genes was controlled via other TFs. Data indicated that the activity of C16:0 was via one or more TF (in black) controlling the expression of the majority of the genes affected by C16:0 in our experiment that were independent from PPARγ (including *SREBF1*, which expression was only downregulated by the inhibition of PPARγ but was still strongly upregulated compared to control). Expression of several genes, such as *GPAM*, *LPIN1*, and *AGPAT6*, was controlled by C16:0 via a TF which activity was somewhat inhibited by the activity of PPARγ (in orange). C18:0 controlled the expression of 9 genes via PPARγ, including *LPL* which effect is only numerical and only partly via this TF. The expression of other genes was controlled via other TFs. Data indicated that the activity of C18:0 was via one or more TF (in black) independent from PPARγ controlling the expression of only 2 measured genes (*INSIG1* and *SREBF1*). Expression of several genes (*ACSL1*, *AGPAT6*, *MID1IP1*, *PPARG*, and *NCOR1*) was upregulated by C18:0 via a TF which activity was inhibited by the activity of PPARγ (in orange).

**Figure 6 vetsci-06-00054-f006:**
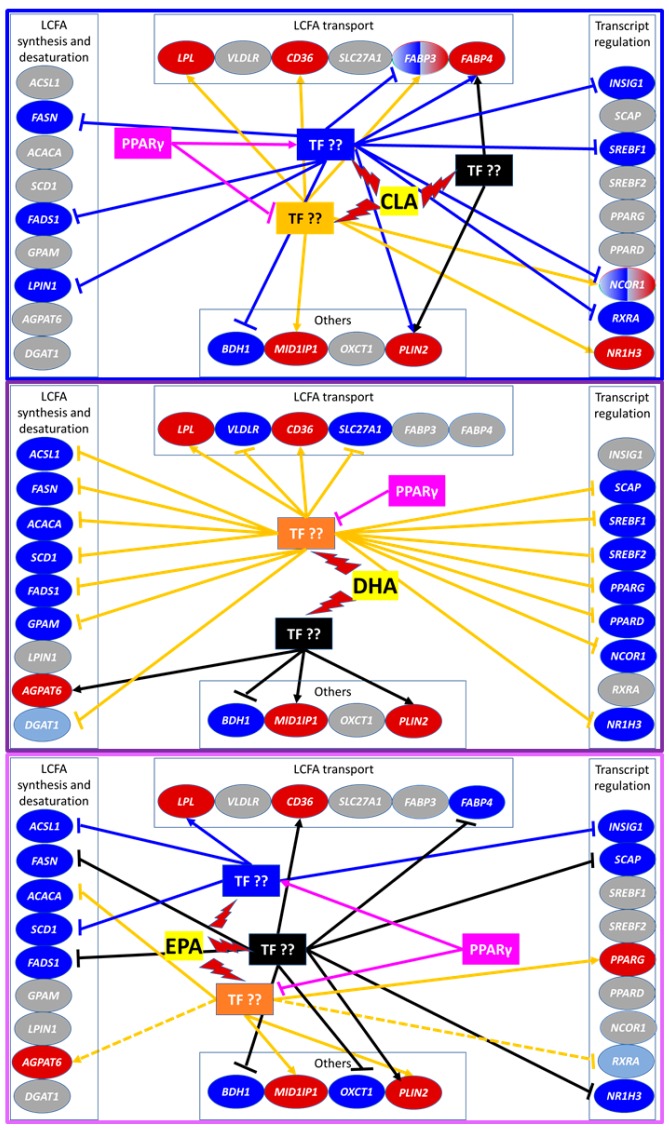
Schematic summary of the results from the present experiment for the response of expression of various genes in goat mammary epithelial cells to unsaturated fatty acids *t10,c12*-CLA, DHA, and EPA. Genes are clustered in functional groups related to milk fat synthesis. As reported, it is also the possible interaction of LCFA with PPARγ or other unidentified transcription factors (TF) to elicit the observed responses. Grey shade denotes no significant change, red shade denotes upregulation, and blue shade denotes downregulation relative to control. In connecting lines, arrow head denotes induction while flat head denotes inhibition. Lightning symbol denotes induction of TF by the LCFA. *t10,c12*-CLA decreased the transcription of 9 genes via a TF (in blue) which activity was induced by activation of PPARγ. The increase in transcription of 6 genes (i.e., *LPL*, *CD36*, *FABP3*, *AGPAT6*, *MID1IP1*, and *NCOR1*) by *t10,c12*-CLA was via a TF (in orange) which activity was inhibited by PPARγ. The transcription of only two of the measured genes (*FABP4* and *PLIN2*) was increased by *t10,c12*-CLA acting upon an unknown TF (in black) that was independent from PPARγ. DHA decreased the transcription of 16 genes and increased the transcription of 2 genes (*LPL* and *CD36*) via a TF (in orange) which activity was inhibited by the activation of PPARγ. Transcription of 3 genes (*AGPAT6*, *MID1IP1*, and *PLIN2*) was upregulated and transcription of *BDH* was downregulated by DHA via a TF that was independent from PPARγ (in black). EPA decreased the transcription of 12 genes but none directly via PPARγ. However, the transcription of *AGPAT6* and *PPARG* was upregulated and transcription of *ACACA*, *RXRA*, *MID1IP1*, and *PLIN2* was downregulated by EPA via a TF which activity was inhibited by PPARγ (in orange). The transcription of *ACSL1*, *SCD1*, and *INSIG1* was downregulated and the transcription of *LPL* was upregulated by EPA through one or more TF that were dependent from an activation of PPARγ (in blue). Transcription of 9 genes was decreased or increased by EPA via one of more TF (in black) that were independent from PPARγ.

**Table 1 vetsci-06-00054-t001:** Statistical effect of 12 h treatment of primary goat mammary epithelial cells with a PPARγ agonist (i.e., 50 μM Rosiglitazone) and antagonist (i.e., 50 μM GW9662) on transcription of genes related to milk fat synthesis.

*Transcript*	*PPARγ **		
	Activator	Inhibitor	Target Gene
***LCFA Synthesis and Desaturation and TAG Synthesis***			
*ACACA*	No	↓ (*p* = 0.09)	Maybe
*ACSL1*	No	↓	Likely
*AGPAT6*	↑	No	Likely not ^#^
*GPAM*	No	↓	Likely
*DGAT1*	No	↓ (*p* = 0.06)	Maybe
*FADS1*	No	No	No
*FASN*	↑	↑	??
*LPIN1*	No	↓	Likely
*SCD1*	↑	No	Yes
***LCFA Transport***			
*CD36*	No	No	No
*FABP3*	No	↓	Likely
*FABP4*	No	No	No
*LPL*	No	↓	Likely
*SLC27A1*	No	↓	Likely
*VLDLR*	No	No	No
***Transcription Regulation***			
*INSIG1*	No	No	No
*NCOR1*	No	↓	Likely
*NR1H3*	↑	↓	Yes
*PPARD*	No	↓	Likely
*PPARG*	No	↓	Likely
*RXRA*	No	No	No
*SCAP*	No	No	No
*SREBF1*	No	No	No
*SREBF2*	No	↓	Likely
***Others***			
*BDH1*	No	No	No
*MID1lP1*	No	No	No
*OXCT1*	No	No	No
*PLIN2*	↑	↑ (*p* = 0.11)	Maybe

* ↑ denotes significant (*p* < 0.05) upregulation; ↓ denotes significant (*p* < 0.05) downregulation. The target genes are indicated as: “Maybe” when the transcription of the gene either tended to be affected by the agonist or antagonist; “likely” denotes transcripts for which the basal expression requires PPARγ activation but Rosiglitazone failed to induce further their transcription; and “yes” denotes transcripts that were significantly affected by the agonist and/or antagonist. ‘??’ represents unknown or unclear.

**Table 2 vetsci-06-00054-t002:** Summary of the effect of 12 h treatment of primary goat mammary epithelial cells with 100 μM long-chain fatty acids in the absence/presence of 50 μM PPARγ inhibitor GW9662. Symbol (↑, ↓, No) = effect compared to the control/Symbol (↑, ↓, No) = effect compared to the PPARγ inhibitor control.

*Transcript*	*Effect of LCFA (without/with GW9662) **				
	C16:0	C18:0	CLA	DHA	EPA
***LCFA Synthesis and Desaturation and TAG Synthesis***					
*ACACA*	**-**/**-**	**-**/**-**	**-**/**-**	**-**/↓	**-**/↓
*ACSL1*	**-**/**-**	**-**/↑	**-**/**-**	**-**/↓	↓/-
*AGPAT6*	**-**/↑	↑/↑	**-**/↑	↑/↑	**-**/↑
*DGAT1*	**-**/**-**	**-**/**-**	**-**/**-**	**-**/↓^#^	**-**/**-**
*FADS1*	↑/↑	↑/-	↓/-	**-**/↓	↓/↓
*FASN*	↑/-	↑/-	**-**/**-**	**-**/↓	↓/↓
*GPAM*	**-**/↓	**-**/**-**	**-**/**-**	**-**/↓	↓/↓
*LPIN1*	**-**/↑	**-**/**-**	↓/-	**-**/**-**	**-**/**-**
*SCD1*	↑/↑	↑/-	**-**/**-**	**-**/↓^$^	↓/-
***LCFA Transport***					
*CD36*	↑/↑	↑/-	↑/-	**-**/↑	**-**/↑
*FABP3*	↑/↑	↑/-	↓/↑	**-**/**-**	**-**/**-**
*FABP4*	↑/↑	↑/-	↑/↑	**-**/**-**	↓/-
*LPL*	**↑****↑**/↑	**-**/↑	**-**/↑	**-**/↑	**-**/↑
*SLC27A1*	**-**/**-**	**-**/**-**	**-**/**-**	**-**/↓^$^	**-**/**-**
*VLDLR*	**-**/**-**	**-**/**-**	**-**/**-**	**-**/↓^$^	**-**/**-**
***Transcription Regulation***					
*INSIG1*	↑/↑	↑/↑	↓/-	**-**/**-**	↓/-
*NCOR1*	↓/-	↓/↑	↓/↑	**-**/↓^$^	**-**/**-**
*NR1H3*	**-**/**-**	**-**/**-**	**-**/**-**	**-**/↓	↓/↓
*PPARD*	**-**/**-**	**-**/**-**	**-**/**-**	**-**/↓	**-**/**-**
*PPARG*	↑/↑	**-**/↑	**-**/**-**	**-**/↓	**-**/↑
*RXRA*	**-**/**-**	**-**/**-**	↓/-	**-**/**-**	**-**/↓^#^
*SCAP*	**-**/**-**	**-**/**-**	**-**/**-**	**-**/↓	↓/↓
*SREBF1*	**↑****↑**/↑	↑/↑	↓/-	**-**/↓	**-**/**-**
*SREBF2*	**-**/**-**	**-**/**-**	**-**/**-**	**-**/↓^$^	**-**/**-**
***Others***					
*BDH1*	**-**/**-**	**-**/**-**	↓/-	↓/↓	↓/↓
*MID1lP1*	↑/↑	**-**/↑	**-**/↑	**-**/↑	**-**/↑
*OXCT1*	**-**/**-**	**-**/**-**	**-**/**-**	**-**/**-**	↓/↓
*PLIN2*	↑/↑	↑/-	↑/↑	↑/↑	↑/**↑****↑**
***PPARγ agonist***	**Weak**	**Weak+**	**No**	**No**	**No**

* “**-**” denotes no effect; ↑ denotes upregulation; ↓ denotes downregulation. ↑↑ denotes further upregulation compared to the other condition (i.e., with vs. without PPARγ inhibitor).

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
