# Peer review of "Nutrigenomic Effect of Saturated and Unsaturated Long Chain Fatty Acids on Lipid-Related Genes in Goat Mammary Epithelial Cells: What Is the Role of PPARγ?"

_vetsci, 2019, doi:10.3390/vetsci6020054_

Round 1

Reviewer 1 Report

The authors found the potential function of PPARgamma on the regulation of lipid-related genes in goat mammary epithelial cells in response to saturated and unsaturated 3 long chain fatty acids treatment. There are some concerns should be address before acceptance.

1. PPAR family function on gene transcription via its protein binding to promoter of target genes. PPARgamma protein level should be measured. 

2. Lipogenesis change would affect TG content, this must be shown here.

3. GW9662, will inhibit PPARα and σ activity at a high concentration, it's hard to make a conclusion that only PPARγ function on lipid-related gene expressions using 50 μM. 

Author Response

The authors found the potential function of PPARgamma on the regulation of lipid-related genes in goat mammary epithelial cells in response to saturated and unsaturated 3 long chain fatty acids treatment. There are some concerns should be address before acceptance.

AU: thanks for reviewing our manuscript!

1. PPAR family function on gene transcription via its protein binding to promoter of target genes. PPARgamma protein level should be measured. 

AU: Protein expression of PPARgamma have been previously determined in the same GMEC. We have added a reference to support this point.

2. Lipogenesis change would affect TG content, this must be shown here.

AU: although we agree with the reviewer that activation of PPAR would affect lipogenesis the purpose of our work was strictly nutrigenomics (i.e,  effect of LCFA on transcription of specific genes). Certainly measuring TG would have helped interpretation of the biological effect of LCFA; however, those have been somewhat well-defined by prior work, some cited in our manuscript. Thus, we do not think it is strictly necessary to measure TG in order to determine is a LCFA affect the transcription of target genes.

3. GW9662, will inhibit PPARα and σ activity at a high concentration, it's hard to make a conclusion that only PPARγ function on lipid-related gene expressions using 50 μM. 

AU: we agree with the reviewer that this is a major limitation in interpreting our data. We have already indicated this in the manuscript in the section 4.1; however, we have now added a “Limitations” section to highlight this and other limitations.

Reviewer 2 Report

Vargas-Bello-Pérez et al. present a study investigating transcriptional regulation of milk fat-related genes and the impact of PPARg. The study applies multiple types of fatty acids as well as agonist and antagonist of PPARg in order to track transcriptional changes in GMEC cell line. The rationale behind the experimental design is logical and clear. The novelty of the results concerns mostly the cell line used and comparison between bovine and goat cells. The study fails at delivering sufficient amount of data, proper study design or statistical analysis.

Major criticism

The authors present data from only one experimentand of weak quality. The presented amount of the data is minimal and does not seem to be sufficient for standards of publication in an international journal.

The presented data were generated by an experiment with three technical replicates. Any guidelines for cell culture experiment state that this is not enough. The cells need to be from different passages, optimally different frozen samples to be considered biological replicates. Each biological replicate group should contain several technical replicates. Therefore, the sample size is too small to make any statements based on the results. 

Statistic is a major problem in the manuscript. In the materials and methods section the authors state “Significance was declared with a P-value£0.05 and tendency with a P-value between 0.05 and 0.10.” Therefore, the authors state p value bellow 0,05 as statistically significant and it seems that they do not perform correction for multiple testing. Even though they test statistical significance for 14 groups. Therefore, 0,05 should not be considered significance threshold and “trends” should definitely not be mentioned. Moreover, in the text authors just state differences in gene regulation for p values 0,1-0,05 without using the word “trend” as if it is statistically significant. Even more, in the text phrases like “Numerically downregulated” are used for p values higher than 0,1! 

Table 1 column “Target gene” describes speculation presented as proven fact. To describe gene as a target more advanced experiments are required. Besides, the table is not a result but summary of results and as such should be in the discussion part.

For the quantity and quality of the resented results the “Result” section is much too long. 

Discussion is very speculative, drifts far from the presented results and is too long.

Legends (at least of the first figure) should explain the meaning of the names of the experimental groups and abbreviations. The system used for highlining statistical results is very confusing.

In my opinion the above reasons completely disqualify the study.

Author Response

Vargas-Bello-Pérez et al. present a study investigating transcriptional regulation of milk fat-related genes and the impact of PPARg. The study applies multiple types of fatty acids as well as agonist and antagonist of PPARg in order to track transcriptional changes in GMEC cell line. The rationale behind the experimental design is logical and clear. The novelty of the results concerns mostly the cell line used and comparison between bovine and goat cells. The study fails at delivering sufficient amount of data, proper study design or statistical analysis.

AU: We thank the reviewer for the time committed to review our manuscript and for the comments provided.

Major criticism

The authors present data from only one experiment and of weak quality. The presented amount of the data is minimal and does not seem to be sufficient for standards of publication in an international journal.

The presented data were generated by an experiment with three technical replicates. Any guidelines for cell culture experiment state that this is not enough. The cells need to be from different passages, optimally different frozen samples to be considered biological replicates. Each biological replicate group should contain several technical replicates. Therefore, the sample size is too small to make any statements based on the results.

AU: Although we agree with the reviewer that “biological replicates” would have strengthen the results (we think that repeating the same experiment with the use of different passage cells would increase the statistical power, but not the relevance of the study, especially for primary mammary epithelial cells) we disagree with the conclusion that the data are weak. When few replicate are used the risk is to not find any statistical effect; thus, the effect that was determined to be statistically significant should be considered as such.

Statistic is a major problem in the manuscript. In the materials and methods section the authors state “Significance was declared with a P-value£0.05 and tendency with a P-value between 0.05 and 0.10.” Therefore, the authors state p value bellow 0,05 as statistically significant and it seems that they do not perform correction for multiple testing. Even though they test statistical significance for 14 groups. Therefore, 0,05 should not be considered significance threshold and “trends” should definitely not be mentioned. Moreover, in the text authors just state differences in gene regulation for p values 0,1-0,05 without using the word “trend” as if it is statistically significant. Even more, in the text phrases like “Numerically downregulated” are used for p values higher than 0,1! 

AU: We thanks the reviewer for pointing out the wrong interpretation of the tendency. We have checked the manuscript and remove any discussion about effect with P>0.10. For multiple comparison. We agree that multiple comparison adjustment generally increases the robustness of conclusions; however, there are not 14 comparisons as indicated, but only a maximum of 6. As indicated in the statistical section, data were analyzed separately for the effect of each agonist, antagonist, and LCFA; a second analysis was done to assess the effect of the antagonist + LCFA, and a third analysis to assess the comparison LCFA and antagonist+LCFA (the latter has not multiple comparison). Thus, the use of multiple comparison adjustment would appear excessive.

Table 1 column “Target gene” describes speculation presented as proven fact. To describe gene as a target more advanced experiments are required. Besides, the table is not a result but summary of results and as such should be in the discussion part.

AU: we thank the reviewer for pointing this out. We agree that additional molecular techniques should be used to prove that the one indicated as targets are indeed PPAR target genes. We addressed the concern of the reviewer in the limitation section now added into the manuscript.

For the quantity and quality of the resented results the “Result” section is much too long. 

AU: Although we agree that the result section is long, we think that it would be unfair to readers to reduce this section. The use of a combination of antagonist and LCFA with the large number of measured genes makes the interpretation of the data difficult. Thus, a detailed description of results will help interpreting the data correctly.

Discussion is very speculative, drifts far from the presented results and is too long.

AU: We respectfully disagree with the reviewer. Although, our data have limitations as now clearly indicated in the section “Limitations” we think that the discussion section correctly tackles the four main findings from our data as indicated by the title of the 4 sections of the discussion. Unclear where the reviewer sees that the discussion drift far from the results, considering that the discussion is concentrated in distilling the main findings from the data. In order to decrease the length of the discussion and account for a point raised by another reviewer we have removed the introductory section of the discussion and integrated it into the introduction. We have revisited the discussion to minimize or eliminate unnecessary speculations.

Legends (at least of the first figure) should explain the meaning of the names of the experimental groups and abbreviations. The system used for highlining statistical results is very confusing.

AU: we have improved the legend as indicated

In my opinion the above reasons completely disqualify the study.

Reviewer 3 Report

General comments:

The manuscript by Vargas-Bello-Pérez focused on effects of saturated and unsaturated long chain fatty acids on lipid metabolism related genes in goat mammary epithelial cells (GMEC). Two saturated LCFA (C16 and C18) and three unsaturated LCFA (CLA, DHA, and EPA) were used as treatments in GMEC. Agonist and antagonist of PPARγ were also used in the treatments with/without LCFA. Seven treatment groups were used for comparisons on three levels (1. Effects of PPARγ agonist and five LCFAs; 2. Effects of each combination of LCFA with GW9662; 3. Compare each LCFA with/without GW9662), which made the experimental design complicated and confusing. Results are about alterations of genes associated with lipid metabolism including fatty acid synthesis and desaturation, fatty acid transport, lipogenesis, lipid droplet formation and ketone body utilization. The authors also compared GMEC data with MACT cells. The major limitation of the study is that the rationale of experimental design is unclear. More details are needed in the introduction section though authors did discuss in the beginning of discussion part. Data were solely based on cell culture and it is difficult to connect gene expression levels in GMEC with real biology story. The discussion is not thorough and it is hard for general readers to catch up the major deliverables. The discussion was mainly based on speculations and no supporting data besides transcriptomics. 

Specific comments:

Line (L) 41-67: The rationale of study is unclear. Please add the description of five LCFAs and their potential roles. 

L81: Intra- and inter- CVs are needed.

L82-88: How did you determine the amount of treatments (e.g., 50 μm and 100 μm). 

L136: In ‘Results’, when talking about genes, you need to explain which pathways they belong too. Though the Table S2 includes all information, corresponding pathways should be mentioned for each gene or gene group. For example, genes in lines 154-155 are related with lipogenesis. 

L148-151: Why FASN was increased under both PPARγ agonist and antagonist?

L156-157: Where is the data? 

L188-193: The definition of ‘Maybe’, ‘Likely’, ‘Yes’, and ‘No’ is confusing.

L371: The discussion is mainly based on speculations. What are the main differences between saturated and unsaturated LCFAs? Why they are different? How they can directly affect lipid-related genes?

Author Response

The manuscript by Vargas-Bello-Pérez focused on effects of saturated and unsaturated long chain fatty acids on lipid metabolism related genes in goat mammary epithelial cells (GMEC). Two saturated LCFA (C16 and C18) and three unsaturated LCFA (CLA, DHA, and EPA) were used as treatments in GMEC. Agonist and antagonist of PPARγ were also used in the treatments with/without LCFA. Seven treatment groups were used for comparisons on three levels (1. Effects of PPARγ agonist and five LCFAs; 2. Effects of each combination of LCFA with GW9662; 3. Compare each LCFA with/without GW9662), which made the experimental design complicated and confusing. Results are about alterations of genes associated with lipid metabolism including fatty acid synthesis and desaturation, fatty acid transport, lipogenesis, lipid droplet formation and ketone body utilization. The authors also compared GMEC data with MACT cells. The major limitation of the study is that the rationale of experimental design is unclear. More details are needed in the introduction section though authors did discuss in the beginning of discussion part. Data were solely based on cell culture and it is difficult to connect gene expression levels in GMEC with real biology story. The discussion is not thorough and it is hard for general readers to catch up the major deliverables. The discussion was mainly based on speculations and no supporting data besides transcriptomics. 

AU: we thank the reviewer for spending time to review our manuscript and for specific comments that helped to address the concerns. We have now provided more details in the introduction about the chosen experimental design. We agree that it is difficult to speculate about real biological effects of our findings; however, we struggle to see where the interpretation of the data is speculative. We have pointed out four major findings (as provided in the title of the four sections of the discussion) that appears from the data. We revised the discussion to minimize possible speculations. In order to account for several limitations indicated by all reviewers we have added a section titled “Limitations” in the discussion

Specific comments:

Line (L) 41-67: The rationale of study is unclear. Please add the description of five LCFAs and their potential roles. 

AU: addressed as indicated.

L81: Intra- and inter- CVs are needed.

AU: addressed as indicated

L82-88: How did you determine the amount of treatments (e.g., 50 μm and 100 μm). 

AU: addressed

L136: In ‘Results’, when talking about genes, you need to explain which pathways they belong too. Though the Table S2 includes all information, corresponding pathways should be mentioned for each gene or gene group. For example, genes in lines 154-155 are related with lipogenesis.

AU: addressed. Divided the results in sections indicating the main functions of genes.

L148-151: Why FASN was increased under both PPARγ agonist and antagonist?

AU: good question….we have no idea why that happened. As indicated in the limitation section, use of other molecular approaches may help to interpret those data

L156-157: Where is the data? 

AU: addressed. It is Figure 4

L188-193: The definition of ‘Maybe’, ‘Likely’, ‘Yes’, and ‘No’ is confusing.

AU: Thanks for point that out. We have now (hopefully) clarified it

L371: The discussion is mainly based on speculations. What are the main differences between saturated and unsaturated LCFAs? Why they are different? How they can directly affect lipid-related genes?

AU: As discussed above we have tried to minimize speculations; however, as indicated by the data, the differences between the saturated and unsaturated LCFA is tackled in the discussion and summarize in the conclusion. Of course, the conclusion is mostly about PPAR, but we also indicated that the differences are likely due to modulation of different transcription factors. Not sure how we can say more without the risk of excessive speculations.

Round 2

Reviewer 1 Report

Accept the revised version.

Reviewer 2 Report

The authors improved the manuscript and provided the missing information. There seems to be a difference in the judgment of proper experimental design and statistical analysis between the authors and the reviewer. In my opinion, the study still does not provide sufficient evidence to support the described claims.

Reviewer 3 Report

The authors addressed my comments carefully and they did a good job in convincing me about the significance of current study. They provided more details about the rationale of experimental design, which made it much easier to follow and understand. Nothing is perfect, however, the authors could list the limitations of the study. The current version of manuscript is of high quality and it fits well to the journal Veterinary Sciences. I do not have more comments.